# ENTROPY-SGD OPTIMIZES THE PRIOR OF A PAC-BAYES BOUND: DATA-DEPENDENT PAC-BAYES PRIORS VIA DIFFERENTIAL PRIVACY

## ABSTRACT

We show that Entropy-SGD (Chaudhari et al., 2017), when viewed as a learning algorithm, optimizes a PAC-Bayes bound on the risk of a Gibbs (posterior) classifier, i.e., a randomized classifier obtained by a risk-sensitive perturbation of the weights of a learned classifier. Entropy-SGD works by optimizing the bound's prior, violating the hypothesis of the PAC-Bayes theorem that the prior is chosen independently of the data. Indeed, available implementations of Entropy-SGD rapidly obtain zero training error on random labels and the same holds of the Gibbs posterior. In order to obtain a valid generalization bound, we show that an $\varepsilon$-differentially private prior yields a valid PAC-Bayes bound, a straightforward consequence of results connecting generalization with differential privacy. Using stochastic gradient Langevin dynamics (SGLD) to approximate the well-known exponential release mechanism, we observe that generalization error on MNIST (measured on held out data) falls within the (empirically nonvacuous) bounds computed under the assumption that SGLD produces perfect samples. In particular, Entropy-SGLD can be configured to yield relatively tight generalization bounds and still fit real labels, although these same settings do not obtain state-of-the-art performance.

## 1 INTRODUCTION

Optimization is central to much of machine learning, but generalization is the ultimate goal. Despite this, the generalization properties of many optimization-based learning algorithms are poorly understood. The standard example is stochastic gradient descent (SGD), one of the workhorses of deep learning, which has good generalization performance in many settings, even under overparametrization (Neyshabur, Tomioka, and Srebro, 2014), but rapidly overfits in others (Zhang et al., 2017). Can we develop high performance learning algorithms with provably strong generalization guarantees? Or is their a limit?

In this work, we study an optimization algorithm called Entropy-SGD (Chaudhari et al., 2017), which was designed to outperform SGD in terms of generalization error when optimizing an empirical risk. Entropy-SGD minimizes an objective $f : \mathbb{R}^p \to \mathbb{R}$ indirectly by performing (approximate) stochastic gradient ascent on the so-called local entropy

$$F(\mathbf{w}) \stackrel{\text{def}}{=} C + \log \mathbb{E}_{\xi \sim \mathcal{N}}[e^{-f(\mathbf{w}+\xi)}] \stackrel{\text{def}}{=} C + \log \int \exp(-f(\mathbf{w}+x)) \mathcal{N}(\mathrm{d}x), \tag{1}$$

where $C$ is a constant and $\mathcal{N}$ denotes a zero-mean isotropic multivariate normal distribution on $\mathbb{R}^p$.

Our first contribution is connecting Entropy-SGD to results in statistical learning theory, showing that maximizing the local entropy corresponds to minimizing a PAC-Bayes bound (McAllester, 1999) on the risk of the so-called Gibbs posterior. The distribution of $\mathbf{w} + \xi$ is the PAC-Bayesian "prior", and so optimizing the local entropy optimizes the bound's prior. This connection between local entropy and PAC-Bayes follows from a result due to Catoni (2007, Lem. 1.1.3) in the case of bounded risk. (See Theorem 4.1.) In the special case where $f$ is the empirical cross entropy, the local entropy is literally a Bayesian log marginal density. The connection between minimizing PAC-Bayes bounds under log loss and maximizing log marginal densities is the subject of recent work

by Germain et al. (2016). Similar connections have been made by Zhang (2006a); Zhang (2006b); Grünwald (2012); Grünwald and Mehta (2016).

Despite the connection to PAC-Bayes, as well as theoretical results by Chaudhari et al. suggesting that Entropy-SGD may be more stable than SGD, we demonstrate that Entropy-SGD (and its corresponding Gibbs posterior) can rapidly overfit, just like SGD. We identify two changes, motivated by theoretical analysis, that suffice to control generalization error, and thus prevent overfitting.

The first change relates to the stability of optimizing the prior mean. The PAC-Bayes theorem requires that the prior be independent of the data, and so by optimizing the prior mean, Entropy-SGD invalidates the bound. Indeed, the bound does not hold empirically. While a PAC-Bayes prior may not be chosen based on the data, it can depend on the data distribution. This suggests that if the prior depends only weakly on the data, it may be possible to derive a valid bound.

We formalize this intuition using differential privacy (Dwork, 2006; Dwork et al., 2015b). By modifying the cross entropy loss to be bounded and replacing SGD with stochastic gradient Langevin dynamics (SGLD; Welling and Teh, 2011), the data-dependent prior mean can be shown to be $(\varepsilon, \delta)$-differentially private (Wang, Fienberg, and Smola, 2015; Minami et al., 2016). We refer to the SGLD variant as Entropy-SGLD. Using results connecting statistical validity and differential privacy (Dwork et al., 2015b, Thm. 11), we show that an $\varepsilon$-differentially private prior mean yields a valid, though looser, generalization bound using the PAC-Bayes theorem. (See Theorem 5.4.)

A gap remains between pure and approximate differential privacy. Under some technical conditions, in the limit as the number of iterations diverges, the distribution of SGLD's output is known to converge weakly to the corresponding stationary distribution, which is the well-known exponential mechanism in differential privacy (Teh, Thiery, and Vollmer, 2016, Thm. 7). Weak convergence, however, falls short of implying that SGLD achieves pure $\varepsilon$-differential privacy. We proceed under the approximation that SGLD enjoys the same privacy as the exponential release mechanism, and apply our $\varepsilon$-differentially private PAC-Bayes bound. We find that the corresponding 95% confidence intervals are reasonably tight but still conservative in our experiments. While the validity of our bounds are subject to our approximation, the bounds give us a view as to the limitations of combining differential privacy with PAC-Bayes bounds: when the privacy of Entropy-SGLD is tuned to contribute no more than $2\varepsilon^2 \times 100 \approx 0.2\%$ to the generalization error, the test error of the learned network is 3–8%, which is approximately 5–10 times higher than the state of the art, which for MNIST is between 0.2-1%, although the community has almost certainly overfit its networks/learning rates/loss functions/optimizers to MNIST. We return to these points in the discussion.

The second change pertains to the stability of the stochastic gradient estimate made on each iteration of Entropy-SGD. This estimate is made using SGLD. (Hence Entropy-SGD is SGLD within SGD.) Chaudhari et al. make a subtle but critical modification to the noise term in SGLD update: the noise is divided by a factor that ranges from $10^3$ to $10^4$. (This factor was ostensibly tuned to produce good empirical results.) Our analysis shows that, as a result of this modification, the Lipschitz constant of the objective function is approximately $10^6$–$10^8$ times larger, and the conclusion that the Entropy-SGD objective is smoother than the original risk surface no longer stands. This change to the noise also negatively impacts the differential privacy of the prior mean. Working backwards from the desire to obtain tight generalization bounds, we are led to divide the SGLD noise by a factor of only $\sqrt[4]{m}$, where $m$ is the number of data points. (For MNIST, $\sqrt[4]{m} \approx 16$.) The resulting bounds are nonvacuous and tighter than those recently published by Dziugaite and Roy (2017), although it must be emphasized that the bound presented here hold subject to the approximation concerning privacy of the prior mean, which is certainly violated but to an unknown degree.

We begin with a review of some related work, before introducing sufficient background so that we can make a formal connection between local entropy and PAC-Bayes bounds. We then introduce a differentially private PAC-Bayes bound. In Section 6, we present experiments on MNIST which provide evidence for our theoretical analysis. (Empirical validation is required in order to address the aforementioned gap between pure and approximate differential privacy.) We close with a short discussion.

## 2 RELATED WORK

This work was inspired in part by Zhang et al. (2017), who highlight empirical properties of SGD that were not widely appreciated within the theory community, and propose a simple linear model to explain the phenomenon. They observe that, without regularization, SGD can achieve zero training error on MNIST and CIFAR, *even if the labels are chosen uniformly at random.* At the same time, SGD obtains weights with very small generalization error with the original labels. The first observation is strong evidence that the set of classifier accessible to SGD within a reasonable number of iterations is extremely rich. Indeed, with probability almost indistinguishable from one, fitting random labels on a large data set implies that the Rademacher complexity of this effective hypothesis class is essentially the maximum possible (Bartlett and Mendelson, 2003, Thm. 11).

The second observation suggests that SGD is performing some sort of capacity control. Zhang et al. show that SGD obtains the minimum norm solution for a linear model, and thus performs implicit regularization. They suggest a similar phenomenon may occur when using SGD to training neural networks. Indeed, earlier work by Neyshabur, Tomioka, and Srebro (2014) observed similar phenomena and argued for the same point: implicit regularization underlies the ability of SGD to generalize, even under massive overparametrization. Subsequent work by Neyshabur, Tomioka, and Srebro (2015) introduced "path" norms as a better measure of the complexity of ReLU networks. Despite progress, these new norms have not yet lead to nonvacuous generalization bounds (Dziugaite and Roy, 2017, App. D).

There has been recent progress: Dziugaite and Roy (2017) describe PAC-Bayes bounds, built by perturbing the weights learned by SGD. (The authors were motivated in part by Entropy-SGD and empirical findings relating to "flat minima".) Their bounds are controlled by 1) the "flatness" of empirical risk surface near the SGD solution and 2) the L2 distance between the learned weights and the random initialization. The bounds are also found to be numerically nonvacuous. (We return to this aspect below.) Similar bounds are studied in further depth by Neyshabur et al. (2017b). Recent advances have also identified new spectral norm bounds that correlate closely with generalization error and distinguish between true and random labels (Bartlett, Foster, and Telgarsky, 2017; Neyshabur et al., 2017a).

Our work and Entropy-SGD both connect to early work by Hinton and Camp (1993) and Hochreiter and Schmidhuber (1997), which introduced regularization schemes based on information-theoretic principles. These ideas, now referred to as "flat minima", were related to minimizing PAC-Bayes bounds by Dziugaite and Roy (2017), although these bounds are minimized with respect to the posterior, not the prior, as is done by Entropy-SGD. Achille and Soatto (2017) provide an information-theoretic argument for a generalization of the objective of Hinton and Camp. Their objective takes the form of regularized empirical cross entropy

$$\hat{R}_S(Q) + \beta \mathrm{KL}(Q||P), \tag{2}$$

where $Q$ and $P$ are the prior and posterior on the weights, respectively. For an appropriate range of $\beta$, linear PAC-Bayes bounds are exactly of this form. In Achille and Soatto (2017) they empirically observe that varying $\beta$ correlates with a degree of overfitting on a random label dataset. Achille and Soatto (2017) also highlight the connections with variational inference (Kingma, Salimans, and Welling, 2015).

Our work also relates to renewed interest in nonvacuous generalization bounds (Langford, 2002; Langford and Caruana, 2002), i.e., bounds on the numerical difference between the unknown classification error and the training error that are (much) tighter than the tautological upper bound of one. Recently, Dziugaite and Roy (2017) demonstrated nonvacuous generalization bounds for random perturbations of SGD solutions using PAC-Bayes bounds for networks with millions of weights. (The algorithm can be viewed as variational dropout (Kingma, Salimans, and Welling, 2015), with a proper data-dependent prior but without local reparametrization.) Their work builds on the core insight demonstrated nearly 15 years ago by Langford and Caruana (2002), who computed nonvacuous bounds for neural networks five orders of magnitude smaller.

A key aspect of our analysis relies on the stability of a data-dependent prior. Stability has long been understood to relate to generalization (Bousquet and Elisseeff, 2002). Our analysis of Entropy-SGLD rests on results in differential privacy (see (Dwork, 2008) for a survey) and its connection to generalization (Dwork et al., 2015b; Dwork et al., 2015a; Bassily et al., 2016; Oneto, Ridella, and

Anguita, 2017), which can be viewed as a particularly stringent notion of stability. Entropy-SGLD is an instance of differentially private empirical risk minimization, which is well studied, both in the abstract (Chaudhuri, Monteleoni, and Sarwate, 2011; Kifer, Smith, and Thakurta, 2012; Bassily, Smith, and Thakurta, 2014) and in the particular setting of private training via SGD (Bassily, Smith, and Thakurta, 2014; Abadi et al., 2016). Our analysis also relates to the differential privacy of Bayesian and Gibbs posteriors, and approximate sampling algorithms (Mir, 2013; Bassily, Smith, and Thakurta, 2014; Dimitrakakis et al., 2014; Wang, Fienberg, and Smola, 2015; Minami et al., 2016).

In effect, our differentially private PAC-Bayes bound uses a data-distribution-dependent prior, which are permitted in the PAC-Bayesian framework. (Priors must be independent of the data sample, however. Differential privacy allows us to extract information about the distribution from a sample while maintaining statistical validity (Dwork et al., 2015b).)

There is a growing body of work in the PAC-Bayes literature on data-distribution-dependent priors. Write $S$ for a data sample and $Q(S)$ for a data-dependent PAC-Bayesian posterior (i.e., $Q : Z^m \to \mathcal{M}_1(\mathbb{R}^p)$ is a fixed learning algorithm for a randomized classifier). Catoni (2007) makes an extensive study of data-distribution-dependent priors of the form $P^* = P^*(Q) \stackrel{\text{def}}{=} \mathbb{E}_{S \sim \mathscr{D}^m}[Q(S)]$. While such priors were known to minimize the KL term in expectation, Catoni was the first to derive PAC-Bayes excess risk bounds using these priors: focusing on Gibbs posteriors $Q(S) = Q_P(S) \stackrel{\text{def}}{=} P_{\exp(-\tau \hat{R}_S)}$ for some fixed measure $P$, Catoni derives bounds on the complexity term $\text{KL}(Q_P(S)||P^*(Q_P))$ that hold uniformly over all possible data distributions $\mathscr{D}$. Catoni calls such priors and bounds "local". Lever, Laviolette, and Shawe-Taylor (2013) extend this approach to generalization bounds and consider both data-independent and data-dependent choices for $P$. In the later case, $P = P(S)$ and the generalization bound uses the local prior $P^*(Q_P) = \mathbb{E}_{S \sim \mathscr{D}^m}[Q_{P(S)}(S)]$. In our work, we make a data-dependent but private choice of the prior $P = \mathscr{P}(S)$, and then use our differentially private PAC-Bayes generalization bound to control the generalization error of the associated Gibbs posterior $Q_P(S)$ in terms of $\text{KL}(Q_P(S)||P)$. We also evaluated differentially private versions of local bounds, where the complexity term is a uniform bound on $\text{KL}(Q_P(S)||P^*(Q_P))$. The bounds were virtually indistinguishable, and so we do not report them here.

## 3 PRELIMINARIES: SUPERVISED LEARNING, ENTROPY-SGD, AND PAC-BAYES

Let $Z$ be a measurable space, let $\mathscr{D}$ be an unknown distribution on $Z$, and consider the batch supervised learning setting under a loss function bounded below: having observed $S \sim \mathscr{D}^m$, i.e., $m$ independent and identically distributed samples from $\mathscr{D}$, we aim to choose a predictor, parameterized by weight vector $\mathbf{w} \in \mathbb{R}^p$, with minimal *risk*

$$R_{\mathscr{D}}(\mathbf{w}) \stackrel{\text{def}}{=} \mathbb{E}_{z \sim \mathscr{D}}(\ell(\mathbf{w}, z)), \tag{3}$$

where $\ell : \mathbb{R}^p \times Z \to \mathbb{R}$ is measurable and bounded below. (We ignore the possibility of constraints on the weight vector for simplicity.) We will also consider randomized predictors, represented by probability measures $Q \in \mathcal{M}_1(\mathbb{R}^p)$ on $\mathbb{R}^p$, whose risks are defined via averaging,

$$[R_{\mathscr{D}}(Q) \stackrel{\text{def}}{=} \mathbb{E}_{\mathbf{w} \sim Q}(R_{\mathscr{D}}(\mathbf{w})) = \mathbb{E}_{z \sim \mathscr{D}}\left(\mathbb{E}_{\mathbf{w} \sim Q}(\ell(\mathbf{w}, z))\right), \tag{4}$$

where the second equality follows from Fubini's theorem and the fact that $\ell$ is bounded below.

Let $S = (z_1, \dots, z_m)$ and let $\hat{\mathscr{D}} \stackrel{\text{def}}{=} \frac{1}{m}\sum_{i=1}^m \delta_{z_i}$ be the empirical distribution. Given a weight distribution $Q$, such as that chosen by a learning algorithm on the basis of data $S$, its *empirical risk*

$$\hat{R}_S(Q) \stackrel{\text{def}}{=} R_{\hat{\mathscr{D}}}(Q) = \frac{1}{m}\sum_{i=1}^m \mathbb{E}_{\mathbf{w} \sim Q}(\ell(\mathbf{w}, z_i)), \tag{5}$$

will be studied as a stand-in for its risk, which we cannot compute. While $\hat{R}_S(Q)$ is easily seen to be an unbiased estimate of $R_{\mathscr{D}}(Q)$ when $Q$ is independent of $S$, our goal is to characterize the (one-sided) *generalization error* $R_{\mathscr{D}}(Q) - \hat{R}_S(Q)$ when $Q$ is random and dependent on $S$.

One of our focuses will be on classification, where $Z = X \times K$, with $K$ a finite set of classes/labels. A product measurable (in practice, continuous) function $f : \mathbb{R}^p \times X \to K$ maps weight vectors $\mathbf{w}$

to classifiers $f(\mathbf{w},\cdot): X \to K$. The loss function is given by $\ell(\mathbf{w},(x,y)) = g(f(\mathbf{w},x),y)$ for some $g: K \times K \to \mathbb{R}$. In this setting, 0–1 loss corresponds to $g(y',y) = 1$ if and only if $y' \neq y$. In binary classification, we take $K = \{0,1\}$.

We will also consider parametric families of probability-distribution-valued classifiers $f: \mathbb{R}^p \times X \to [0,1]^K$. For every input $x \in X$, the output $f(\mathbf{w},x)$ specifies a probability distribution on K. In this setting, $\ell(\mathbf{w},(x,y)) = g(f(\mathbf{w},x),y)$ for some $g: [0,1]^K \times K \to \mathbb{R}$. The standard loss is then the cross entropy, given by $g((p_1,\ldots,p_K),y) = -\log p_y$. (Under cross entropy loss, the empirical risk is, up to a multiplicative constant, a negative log likelihood.) In the special case of binary classification, the output can be represented simply by an element of $[0,1]$, i.e., the probability the label is one. The binary cross entropy, $\ell_{\mathrm{BCE}}$, is given by $g(p,y) = -y\log(p) - (1-y)\log(1-p)$. Note that cross entropy loss is merely bounded below. We will consider bounded modifications in Appendix B.2.

We will sometimes refer to elements of $\mathbb{R}^p$ and $\mathcal{M}_1(\mathbb{R}^p)$ as classifiers and randomized classifiers, respectively. Likewise, we will often refer to the (empirical) risk as the (empirical) error.

### 3.1 ENTROPY-SGD

Entropy-SGD is a gradient-based learning algorithm proposed by Chaudhari et al. (2017) as an alternative to stochastic gradient descent on the empirical risk surface $\hat{R}_S$. The authors argue that Entropy-SGD has better generalization performance and provide some empirical evidence. Part of that argument is a theoretical analysis of the smoothness of the local entropy surface that Entropy-SGD optimizes in place of the empirical risk surface, as well as a uniform stability argument that they admit rests on assumptions that are violated, but to a small degree empirically. As we have mentioned in the introduction, Entropy-SGD's modifications to the noise term in SGLD result in much worse smoothness. We will modify Entropy-SGD in order to stabilize its learning and, up to some approximations, provably control overfitting.

Entropy-SGD is stochastic gradient ascent applied to the optimization problem:

$$\arg\max_{\mathbf{w}\in\mathbb{R}^p} F_{\gamma,\tau}(\mathbf{w};S), \quad \text{where } F_{\gamma,\tau}(\mathbf{w};S) = \log \int_{\mathbb{R}^p} \exp\left(-\tau\hat{R}_S(\mathbf{w}') - \tau\frac{\gamma}{2}\|\mathbf{w}'-\mathbf{w}\|_2^2\right) d\mathbf{w}'. \quad (6)$$

The objective $F_{\gamma,\tau}(\cdot;S)$ is known as the *local entropy*, and can be viewed as the log partition function of the unnormalized probability density function

$$\mathbf{w}' \mapsto \exp\left(-\tau\hat{R}_S(\mathbf{w}') - \tau\frac{\gamma}{2}\|\mathbf{w}'-\mathbf{w}\|_2^2\right). \quad (7)$$

(We will denote the corresponding distribution by $G_{\gamma,\tau}^{\mathbf{w},S}$.) Assuming that one can exchange differentiation and integration, it is straightforward to verify that

$$\nabla_\mathbf{w} F_{\gamma,\tau}(\mathbf{w};S) = \mathop{\mathbb{E}}_{\mathbf{w}'\sim G_{\gamma,\tau}^{\mathbf{w},S}}(\tau\gamma(\mathbf{w}-\mathbf{w}')), \quad (8)$$

and then the local entropy $F_{\gamma,\tau}(\cdot;S)$ is even differentiable, even if the empirical risk $\hat{R}_S$ is not. Indeed, Chaudhari et al. show that the local entropy and its derivative are Lipschitz. Chaudhari et al. argue informally that maximizing the local entropy leads to "flat minima" in the empirical risk surface, which several authors (Hinton and Camp, 1993; Hochreiter and Schmidhuber, 1997; Baldassi et al., 2015; Baldassi et al., 2016) have argued is tied to good generalization performance (though none of these papers gives generalization bounds, vacuous or otherwise).[1]

Chaudhari et al. propose a Monte Carlo estimate of the gradient,

$$\nabla_\mathbf{w} F_{\gamma,\tau}(\mathbf{w};S) \approx \tau\gamma(\mathbf{w}-\mu_L), \quad \text{with } \mu_1 = \mathbf{w}_1 \text{ and } \mu_{j+1} = \alpha\mathbf{w}'_j + (1-\alpha)\mu_j, \quad (9)$$

---

[1] The local entropy should not be confused with the smoothed risk surface obtained by convolution with a Gaussian kernel: in that case, every point on this surface represents the average risk of a network obtained by perturbing the network parameters according to a Gaussian distribution. The local entropy also relates to a perturbation, but the perturbation is either accepted or rejected based upon its relative performance (as measured by the exponentiated loss) compared with typical perturbations. Thus the local entropy perturbation concentrates on regions of weight space with low empirical risk, provided they have sufficient probability mass under the distribution of the random perturbation. Section 4 yields further insight into the local entropy function.

---

**Algorithm 1** One step of Entropy-SG(L)D along the local entropy gradient

---

**Input:**
    $\mathbf{w} \in \mathbb{R}^p$                                                         $\triangleright$ Current weight
    $S \in Z^m$                                                             $\triangleright$ Data
    $\ell : \mathbb{R}^p \times Z \to \mathbb{R}$                                                  $\triangleright$ Loss
    $\tau, \gamma, \eta, \eta', L, K$                                               $\triangleright$ Parameters
**Output:** Weight vector $\mathbf{w}$ moved along stochastic gradient
 1: **procedure** ENTROPY-SG(L)D-STEP
 2:      $\mathbf{w}', \mu \leftarrow \mathbf{w}$
 3:      **for** $i \in \{1, ..., L\}$ **do**                                   $\triangleright$ Run SGLD for L iterations.
 4:          $\eta_i' \leftarrow \eta'/i$
 5:          $(z_{j_1}, \ldots, z_{j_K}) \leftarrow$ sample size $K$ minibatch from $S$
 6:          $d\mathbf{w}' \leftarrow \frac{\tau}{K} \sum_{i=1}^{K} \nabla_{\mathbf{w}'} \ell(\mathbf{w}', z_{j_i}) - \gamma \tau(\mathbf{w}' - \mathbf{w})$
 7:          $\mathbf{w}' \leftarrow \mathbf{w}' - \frac{1}{2} \eta_i' d\mathbf{w}' + \sqrt{\eta_i'} N(0, I_p)$
 8:          $\mu \leftarrow (1 - \alpha)\mu + \alpha \mathbf{w}'$
 9:      $\mathbf{w} \leftarrow \mathbf{w} - \frac{1}{2} \eta \tau \gamma(\mathbf{w} - \mu) + \underbrace{\sqrt{\eta/\beta} N(0, I_p)}_{\text{Entropy-SGLD only}}$       $\triangleright$ Step along stochastic local entropy $\nabla$
10:      **return w**

---

where $\mathbf{w}_1', \mathbf{w}_2', \ldots$ are (approximately) i.i.d. samples from $G_{\gamma, \tau}^{\mathbf{w}, S}$ and $\alpha \in (0, 1)$ defines a weighted average. Obtaining samples from $G_{\gamma, \tau}^{\mathbf{w}, S}$ may be difficult when the dimensionality of the weight vector is large. Chaudhari et al. use Stochastic Gradient Langevin Dynamics (SGLD; Welling and Teh, 2011), which generates an exact sample in the limit of infinite computation and requires that the empirical risk be differentiable.[2] The final output of Entropy-SGD is the deterministic predictor corresponding to the final weights $\mathbf{w}^*$ achieved by several epochs of optimization.

Algorithm 1 gives a complete description of the stochastic gradient step performed by Entropy-SGD. If we rescale the learning rate, $\eta' \leftarrow \frac{1}{2} \eta' \tau$, lines 6 and 7 are equivalent to

 6:      $d\mathbf{w}' \leftarrow \frac{1}{K} \sum_{i=1}^{K} \nabla_{\mathbf{w}'} \ell(\mathbf{w}', z_{j_i}) - \gamma(\mathbf{w}' - \mathbf{w})$

 7:      $\mathbf{w}' \leftarrow \mathbf{w}' - \eta_i' d\mathbf{w}' + \sqrt{2\eta_i'/\tau} N(0, I_p)$

Notice that the noise term is multiplied by a factor of $\sqrt{2/\tau}$. This follows from the definition of the local entropy. A multiplicative factor $\varepsilon$—called the "thermal noise", but playing exactly the same role as $\sqrt{2/\tau}$ here—appears in the original description of the Entropy-SGD algorithm given by Chaudhari et al. However, $\varepsilon$ does not appear in the definition of local entropy used in their stability analysis. Our derivations highlights that the scaling the noise term in SGLD update has a profound effect: the thermal noise exponentiates the density that defines the local entropy. The smoothness analysis of Entropy-SGD does not take into consideration the role of $\varepsilon$, which is critical because Chaudhari et al. take $\varepsilon$ to be as small as $10^{-3}$ and $10^{-4}$. Indeed, the conclusion that the local entropy surface is smoother no longer holds. We will see that $\tau$ controls the differential privacy and thus the generalization error of Entropy-SGD.

### 3.2 KL DIVERGENCE AND THE PAC-BAYES THEOREM

Let $Q, P$ be probability measures defined on $\mathbb{R}^p$, assume $Q$ is absolutely continuous with respect to $P$, and write $\frac{dQ}{dP} : \mathbb{R}^p \to \mathbb{R}_+ \cup \{\infty\}$ for some Radon–Nikodym derivative of $Q$ with respect to $P$. Then the Kullback–Liebler divergence (or relative entropy) of $P$ from $Q$ is defined to be

$$\text{KL}(Q||P) \stackrel{\text{def}}{=} \int \log \frac{dQ}{dP} \, dQ. \tag{10}$$

For $p, q \in [0, 1]$, we will abuse notation and define

$$\text{KL}(q||p) \stackrel{\text{def}}{=} \text{KL}(\mathscr{B}(q)||\mathscr{B}(p)) = q \log \frac{q}{p} + (1-q) \log \frac{1-q}{1-p}, \tag{11}$$

---

[2] Chaudhari et al. take $L = 20$ steps of SLGD, using a *constant* step size $\eta_j' = 0.2$ on iteration $j$, and weighting $\alpha = 0.75$. It seems unlikely that these settings produce high quality samples.

where $\mathscr{B}(p)$ denotes the Bernoulli distribution on $\{0,1\}$ with mean $p$.

We now present a PAC-Bayes theorem, first established by McAllester (1999). We focus on the setting of bounding the generalization error of a (randomized) classifier on a finite discrete set of labels $K$. The following variation is due to Langford and Seeger (2001) for 0–1 loss (see also (Langford, 2002) and (Catoni, 2007).)

**Theorem 3.1** (PAC-Bayes (McAllester, 1999; Langford and Seeger, 2001)). *Under 0–1 loss, for every $\delta > 0$, $m \in \mathbb{N}$, distribution $\mathscr{D}$ on $\mathbb{R}^k \times K$, and distribution $P$ on $\mathbb{R}^p$,*

$$\mathop{\mathbb{P}}_{S \sim \mathscr{D}^m}\left( (\forall Q)\ \mathrm{KL}(\hat{R}_S(Q)||R_{\mathscr{D}}(Q)) \leq \frac{\mathrm{KL}(Q||P) + \log \frac{2m}{\delta}}{m-1} \right) \geq 1 - \delta. \tag{12}$$

We will also use the following variation of a PAC-Bayes bound, where we consider any bounded loss function.

**Theorem 3.2** (Linear PAC-Bayes Bound (McAllester, 2013; Catoni, 2007)). *Fix $\lambda > 1/2$ and assume the loss takes values in an interval of length $L_{max}$. For every $\delta > 0$, $m \in \mathbb{N}$, distribution $\mathscr{D}$ on $\mathbb{R}^k \times K$, and distribution $P$ on $\mathbb{R}^p$,*

$$\mathop{\mathbb{P}}_{S \sim \mathscr{D}^m}\left( (\forall Q)\ R_{\mathscr{D}}(Q) \leq \frac{1}{1 - \frac{1}{2\lambda}}\left( \hat{R}_S(Q) + \frac{\lambda L_{max}}{m}(\mathrm{KL}(Q||P) + \log\frac{1}{\delta}) \right) \right) \geq 1 - \delta. \tag{13}$$

We introduce several additional generalization bounds when we introduce differential entropy.

## 4   MAXIMIZING LOCAL ENTROPY MINIMIZES A PAC-BAYES BOUND

We now present our first contribution, a connection between the local entropy and PAC-Bayes bounds. We begin with some notation for Gibbs distributions. For a measure $P$ on $\mathbb{R}^p$ and function $g : \mathbb{R}^p \to \mathbb{R}$, let $P[g]$ denote the expectation $\int g(h)P(\mathrm{d}h)$ and, provided $P[g] < \infty$, let $P_g$ denote the probability measure on $\mathbb{R}^p$, absolutely continuous with respect to $P$, with Radon–Nikodym derivative $\frac{\mathrm{d}P_g}{\mathrm{d}P}(h) = \frac{g(h)}{P[g]}$. A distribution of the form $P_{\exp(-\tau g)}$ is generally referred to as a Gibbs distribution. In the special case where $P$ is a probability measure, we call $P_{\exp(-\tau \hat{R}_S)}$ a "Gibbs posterior".

**Theorem 4.1** (Maximizing local entropy optimizes a PAC-Bayes bound's prior). *Assume the loss function takes values in an interval of length $L_{max}$, let $\tau = \frac{m}{\lambda L_{max}}$ for some $\lambda > 1/2$, and let $P$ be a multivariate normal distribution with mean $\mathbf{w}$ and covariance matrix $(\tau\gamma)^{-1}I_p$. Then maximizing the local entropy $F_{\gamma,\tau}(\mathbf{w}; S)$ with respect to $\mathbf{w}$ is equivalent to minimizing a linear PAC-Bayes bound (Theorem 3.2) on the risk $R_{\mathscr{D}}(G_{\gamma,\tau}^{\mathbf{w},S})$ of the Gibbs posterior $G_{\gamma,\tau}^{\mathbf{w},S} = P_{\exp(-\tau\hat{R}_S)}$, where the bound is optimized with respect to the mean $\mathbf{w}$ of $P$.*

*Proof.* Let $m$, $\delta$, $\mathscr{D}$, and $P$ be as in Theorem 3.1 and let $S \sim \mathscr{D}^m$. The linear PAC-Bayes bound (Theorem 3.2) ensures that for any fixed $\lambda > 1/2$ and bounded loss function, with probability at least $1 - \delta$ over the choice of $S$, the bound

$$\left(1 - \frac{1}{2\lambda}\right)\frac{m}{\lambda L_{\max}}R_{\mathscr{D}}(Q) \leq \frac{m}{\lambda L_{\max}}\hat{R}_S(Q) + \mathrm{KL}(Q||P) + g(\delta). \tag{14}$$

holds for all $Q \in \mathscr{M}_1(\mathbb{R}^p)$. Minimizing the upper bound on the risk $R_{\mathscr{D}}(Q)$ of the randomized classifier $Q$ is equivalent to the program

$$\inf_Q \mathop{\mathbb{E}}_{h \sim Q}\big(r(h)\big) + \mathrm{KL}(Q||P) \tag{15}$$

with $r(h) = \frac{m}{\lambda L_{\max}}\hat{R}_S(h)$. By (Catoni, 2007, Lem. 1.1.3), for all $Q \in \mathscr{M}_1(\mathbb{R}^p)$ with $\mathrm{KL}(Q||P) < \infty$,

$$-\log P[\exp(-r)] = \mathop{\mathbb{E}}_{h \sim Q}\big(r(h)\big) + \mathrm{KL}(Q||P) - \mathrm{KL}(Q||P_{\exp(-r)}). \tag{16}$$

Using Eq. (16), we may reexpress Eq. (15) as

$$\inf_Q \mathrm{KL}(Q||P_{\exp(-r)}) - \log P[\exp(-r)]. \tag{17}$$

By the nonnegativity of the Kullback–Liebler divergence, the infimum is achieved when the KL term is zero, i.e., when $Q = P_{\exp(-r)}$. Then

$$\left(1 - \frac{1}{2\lambda}\right) \frac{m}{\lambda L_{\max}} R_{\mathscr{D}}(P_{\exp(-r)}) \leq -\log P[\exp(-r)] + g(\delta). \tag{18}$$

Finally, it is plain to see that $F_{\gamma,\tau}(\mathbf{w};S) = C + \log P[\exp(-r)]$ when $C = \frac{1}{2}p\log(2\pi(\tau\gamma)^{-1})$ is a constant, $\tau = \frac{m}{\lambda L_{\max}}$, and $P = \mathcal{N}(\mathbf{w},(\tau\gamma)^{-1}I_p)$ is a multivariate normal with mean $\mathbf{w}$ and covariance matrix $(\tau\gamma)^{-1}I$. $\qquad\square$

The analysis falls short when the loss function is unbounded, because the PAC-Bayes bound we have used applies only to bounded loss functions. Germain et al. (2016) described PAC-Bayes generalization bounds for unbounded loss functions. (See Grünwald and Mehta (2016) for related work on excess risk bounds and further references). For their bounds to be evaluated on the negative log likelihood loss, one needs some knowledge of the data distribution in order to approximate certain statistics of the deviation of the empirical risk $\hat{R}_S(w)$ from true risk $R_{\mathscr{D}}(w)$.

## 5 DATA-DEPENDENT PAC-BAYES PRIORS VIA DIFFERENTIAL PRIVACY

Theorem 4.1 reveals that Entropy-SGD is optimizing a PAC-Bayes bound with respect to the prior. As a result, the prior $P$ depends on the sample $S$, and the hypotheses of the PAC-Bayes theorem (Theorem 3.1) are not met. Naively, it would seem that this interpretation of Entropy-SGD cannot explain its ability to generalize. Using tools from differential privacy (Dwork, 2006), we show that if the prior term is optimized in a differentially private way, then a PAC-Bayes theorem still holds, at the cost of a slightly looser bound. We will assume basic familiarity with differential privacy, but give basic definitions and results in Appendix A. We use the notation $\mathscr{A}: Z \rightsquigarrow T$ for a (randomized) algorithm that takes as input an element in $Z$ and produces an output in $T$.

The key result we will employ is due to Dwork et al. (2015b, Thm. 11).

**Theorem 5.1.** *Let $m \in \mathbb{N}$, let $\mathscr{A}: Z^m \rightsquigarrow T$, let $\mathscr{D}$ be a distribution over $Z$, let $\beta \in (0,1)$, and, for each $t \in T$, fix a set $R(t) \subseteq Z^m$ such that $\mathbb{P}_{S \sim \mathscr{D}^m}(S \in R(t)) \leq \beta$. If $\mathscr{A}$ is $\varepsilon$-differentially private for $\varepsilon \leq \sqrt{\ln(1/\beta)/(2m)}$, then $\mathbb{P}_{S \sim \mathscr{D}^m}(S \in R(\mathscr{A}(S))) \leq 3\sqrt{\beta}$.*

Using Theorem 5.1, one can compute tail bounds on the generalization error of fixed classifiers, and then, provided that a classifier is learned from data in a differentially private way, the tail bound holds on the classifier, with less confidence. The following two tail bounds are examples of this idea. The first is a simple variant of (Dwork et al., 2015b, Thm. 9) due to Oneto, Ridella, and Anguita (2017, Lem. 2).

**Theorem 5.2.** *Let $m \in \mathbb{N}$ and let $\mathscr{A}: Z^m \rightsquigarrow \mathbb{R}^p$ be $\varepsilon$-differentially private. Then*

$$\mathop{\mathbb{P}}_{S \sim \mathscr{D}^m}\left(|R_{\mathscr{D}}(\mathscr{A}(S)) - \hat{R}_S(\mathscr{A}(S))| \geq \varepsilon + \sqrt{1/m}\right) \leq 3e^{-m\varepsilon^2}.$$

**Theorem 5.3** ((Oneto, Ridella, and Anguita, 2017, Lem. 3))**.** *Let $m \in \mathbb{N}$ and let $\mathscr{A}: Z^m \rightsquigarrow \mathbb{R}^p$ be $\varepsilon$-differentially private. Then*

$$\mathop{\mathbb{P}}_{S \sim \mathscr{D}^m}\left(|R_{\mathscr{D}}(\mathscr{A}(S)) - \hat{R}_S(\mathscr{A}(S))| \geq \sqrt{6\hat{R}_S(\mathscr{A}(S))(\varepsilon + \sqrt{1/m})} + 6(\varepsilon^2 + 1/m)\right) \leq 3e^{-m\varepsilon^2}.$$

### 5.1 AN $\varepsilon$-DIFFERENTIALLY PRIVATE PAC-BAYES BOUND

The PAC-Bayes theorem allows one to choose the prior based on the data-generating distribution $\mathscr{D}$, but not on the data $S \sim \mathscr{D}^m$. Using differential privacy, we can consider a data-dependent prior $\mathscr{P}(S)$.

**Theorem 5.4.** *Under 0–1 loss, for every $\delta > 0$, $m \in \mathbb{N}$, distribution $\mathscr{D}$ on $\mathbb{R}^k \times K$, and $\varepsilon$-differentially private data-dependent prior $\mathscr{P}: Z^m \rightsquigarrow \mathscr{M}_1(\mathbb{R}^p)$,*

$$\mathop{\mathbb{P}}_{S \sim \mathscr{D}^m}\left((\forall Q)\ \mathrm{KL}(\hat{R}_S(Q)||R_{\mathscr{D}}(Q)) \leq \frac{\mathrm{KL}(Q||\mathscr{P}(S)) + \ln 2m + 2\max\{\ln\frac{3}{\delta},\ m\varepsilon^2\}}{m-1}\right) \geq 1 - \delta. \tag{19}$$

*Proof.* Fix a distribution $\mathscr{D}$ on $\mathbb{R}^k \times K$. For every distribution $P$ on $\mathbb{R}^p$, let

$$R(P) = \left\{ S \in Z^m : (\exists Q) \ \mathrm{KL}(\hat{R}_S(Q)||R_{\mathscr{D}}(Q)) \geq (m-1)^{-1} \big( \mathrm{KL}(Q||P) + \ln 2m + \ln(1/\beta) \big) \right\}. \quad (20)$$

It follows from the PAC-Bayes theorem (Theorem 3.1) that $\mathbb{P}_{S \sim \mathscr{D}^m}(S \in R(P)) \leq \beta$. Theorem 5.1 implies that the bound holds with $P$ replaced by $\mathscr{P}(S)$, provided that we inflate the probability of failure.

In particular, let $\delta = 3\sqrt{\beta}$. Then $\ln(1/\beta) = 2\ln(3/\delta)$. By Theorem 5.1, provided $2m\varepsilon^2 \leq \ln(1/\beta)$, then $\mathbb{P}_{S \sim \mathscr{D}^m}(S \in R(\mathscr{P}(S))) \leq \delta$. It follows that, with probability no more than $\delta$ over $S \sim \mathscr{D}^m$, there exists a distribution $Q$ on $\mathbb{R}^p$ such that

$$\mathrm{KL}(\hat{R}_S(Q)||R_{\mathscr{D}}(Q)) \geq \frac{\mathrm{KL}(Q||\mathscr{P}(S)) + \ln 2m + \max\{2m\varepsilon^2, \ln(1/\beta)\}}{m-1}. \quad (21)$$

The bound stated in Eq. (19) follows immediately. $\qquad \square$

Note that the bound holds for any posterior $Q$, including one obtained by optimizing a *different* PAC-Bayes bound. We have chosen to present a differentially private version of Theorem 3.1 rather than Theorem 3.2, because the former tends to be tighter numerically. Giving a differentially private version of Theorem 3.2, or any other PAC-Bayes bound, should be straightforward: one merely needs to decide how to incorporate the constraint between $\varepsilon$, $\beta$, and $m$ in Theorem 5.1. We have chosen to deal with the constraint via a max operation affecting the width of the confidence interval. Note that, in realistic scenarios, $\delta$ is large enough relative to $\varepsilon$ that an $\varepsilon$-differentially private prior $\mathscr{P}(S)$ contributes $2\varepsilon^2$ to the generalization error. Therefore, $\varepsilon$ must be much less than one to not contribute a nontrivial amount to the generalization error. In order to match the $m^{-1}$ rate by which the KL term decays, one must have $\varepsilon \in O(m^{-1/2})$. Our empirical studies use this rate.

## 5.2 DIFFERENTIALLY PRIVATE DATA-DEPENDENT PRIORS

We have already explained that the weights learned by Entropy-SGD can be viewed as the mean of a data-dependent prior $\mathscr{P}(S)$. By Theorem 5.4 and the fact that post-processing does not decrease privacy, it would suffice to establish that the mean is $\varepsilon$-differentially private in order to obtain a risk bound on the corresponding Gibbs posterior classifier.

Entropy-SGD can be viewed as stochastic gradient ascent on the negative local entropy, but with biased gradient estimates. The bias comes from the use of SGLD to compute the expectation in Eq. (8). Putting aside this issue, existing privacy analyses of SGD worsen after every iteration. For the number of iterations necessary to obtain reasonable weights, known upper bounds on the differential privacy of SGD yield vacuous generalization bounds.

The standard (if idealized) approach for optimizing a data-dependent objective in a private way is to use the exponential mechanism (McSherry and Talwar, 2007). In the context of maximizing the local entropy, the exponential mechanism correspond to sampling exactly from the "local entropy (Gibbs) distribution"

$$P_{\exp(\beta F_{\gamma,\tau}(\cdot;S))}, \quad (22)$$

where $\beta > 0$ and $P$ is some measure on $\mathbb{R}^p$. (It is natural to take $P$ to be Lebesgue measure, or a multivariate normal distribution, which would correspond to L2 regularization of the local entropy.) The following result establishes the privacy of a sample from the local entropy distribution:

**Theorem 5.5.** *Let $\gamma, \tau > 0$, and assume the range of the loss is contained in an interval of length $L_{max}$. One sample from the local entropy distribution $P_{\exp(\beta F_{\gamma,\tau}(\cdot;S))}$, is $\frac{2\beta L_{max}\tau}{m}$-differentially private.*

*Proof.* The result follows immediately from the following two lemmas. $\qquad \square$

**Lemma 5.6** ((McSherry and Talwar, 2007, Thm. 6)). *Let $q : Z^m \times \mathbb{R}^p \to \mathbb{R}$ be measurable, let $P$ be a measure on $\mathbb{R}^p$, let $\beta > 0$, and assume $P[\exp(-\beta q(S, \cdot))] < \infty$ for all $S \in Z^m$. Let $\Delta q \stackrel{def}{=} \sup_{S,S'} \sup_{\mathbf{w} \in \mathbb{R}^p} |q(S, \mathbf{w}) - q(S', \mathbf{w})|$, where the first supremum ranges over pairs $S, S' \in Z^m$ that disagree on no more than one coordinate. Let $\mathscr{A} : Z^m \rightsquigarrow \mathbb{R}^p$, on input $S \in Z^m$, output a sample from the Gibbs distribution $P_{\exp(-\beta q(S, \cdot))}$. Then $\mathscr{A}$ is $2\beta\Delta q$-differentially private.*

**Lemma 5.7.** *Let $F_{\gamma,\tau}(\mathbf{w};S)$ be defined as Eq. (6), assume the range of the loss is contained in an interval of length $L_{max}$, and define $q(S,\mathbf{w}) = -F_{\gamma,\tau}(\mathbf{w};S)$. Then $\Delta q \overset{def}{=} \sup_{S,S'} \sup_{\mathbf{w}\in\mathbb{R}^p} |q(S,h) - q(S',h)| \leq \frac{L_{max}\tau}{m}$.*

*Proof.* The proof essentially mirrors that of (McSherry and Talwar, 2007, Thm. 6). □

There are two obvious obstructions to using the exponential mechanism to pick a prior mean: first, cross-entropy loss can change in an unbounded way when swapping a single data point; second, sampling from the local entropy distribution exactly is hard in general. To sidestep the first obstruction, we modify the underlying cross-entropy loss to be bounded by rescaling the probabilities output by the classifier to be bounded away from zero and one, allowing us to invoke Lemma 5.7. (Details of our modification of the cross entropy are described in Appendix B.2.1.)

There is no simple way to sidestep the second obstruction. Instead, we once again use SGLD to generate an approximate sample from the local entropy distribution. In summary, to optimize the local entropy $F_{\gamma,\tau}(\cdot;S)$ in a private way to obtain the prior mean $\mathbf{w}$, we repeatedly perform the SGLD update

$$\mathbf{w} \leftarrow \mathbf{w} - \tfrac{1}{2}\eta\hat{g}(\mathbf{w}) + \sqrt{\eta/\beta}\,N(0,I_p), \tag{23}$$

where at each round $\hat{g}(\mathbf{w})$ is an estimate of the gradient $\nabla_{\mathbf{w}}F_{\gamma,\tau}(\mathbf{w};S)$. (Recall the identity Eq. (8).) As in Entropy-SGD, we construct biased gradient estimates via an inner loop of SGLD. In summary, the only change to Entropy-SGD is the addition of noise in the outer loop. We call the resulting algorithm Entropy-SGLD. (See Algorithm 1. Note that we take $\beta = 1$ in our experiments.)

There have been a number of privacy analyses of SGLD (Mir, 2013; Bassily, Smith, and Thakurta, 2014; Dimitrakakis et al., 2014; Wang, Fienberg, and Smola, 2015; Minami et al., 2016). Most of these analyses deliver $(\varepsilon,\delta)$-differential privacy, but none of them take advantage of the fact that SGLD mixes in the limit as it converges weakly to the Gibbs distributions, under certain technical conditions (Teh, Thiery, and Vollmer, 2016, Thm. 7). *In our analysis and bound calculations, we therefore make the approximation that SGLD has the same privacy as its limiting invariant measure, the exponential mechanism.* Building a less conservative model of the privacy of SGLD is an open problem. However, by making this approximation, we may see the potential/limits of combining differentially private optimization and PAC-Bayesian bounds. We return to the issues again in light of our empirical findings (Section 6) and in our discussion (Section 7).

## 6 NUMERICAL EVALUATIONS ON MNIST

The generalization bounds that we have devised are data-dependent and so the question of their utility is an empirical one that requires data. In this section, we perform an empirical study of SGD, SGLD, Entropy-SGD, and Entropy-SGLD on the MNIST data set, on both convolutional and fully connected architectures, and compare our generalization bounds to estimates based on held-out data.

Under our privacy approximation, SGLD and Entropy-SGLD are $\varepsilon$-differentially private and we take advantage of this fact to apply differentially private versions of two tail bounds and our PAC-Bayes bound. The degree $\varepsilon$ of privacy is determined by the $\tau$ parameter of the local entropy (C.f. thermal noise $\sqrt{2/\tau}$), and then, in turn, $\varepsilon$ contributes to our bounds on the generalization error. As theory predicts, $\tau$ affects the degree of overfitting empirically, and no bound we compute is violated too frequently. Of course, the validity of our generalization bounds rests on the degree to which our privacy approximation is violated.[3] We reflect on our approximation in light of our empirical results, and then return to this point in the discussion.

The weights learned by SGD, SGLD, and Entropy-SGD are treated differently from those learned by Entropy-SGLD. In the former case, the weights parametrize a neural network as usual, and the training and test error are computed using these weights. In the latter case, the weights are taken to be the mean of a multivariate normal prior, and we evaluate the training and test error of the

---

[3]A number of factors might influence the degree to which our privacy approximation is invalid: step size, batch size, use of batch normalization, number of iterations, use of Metropolis–Hastings corrections. However, when we are operating far from the worst case, we suspect these details are not relevant to observable phenomena, and so our theoretical analysis and empirical findings should serve as a useful guide.

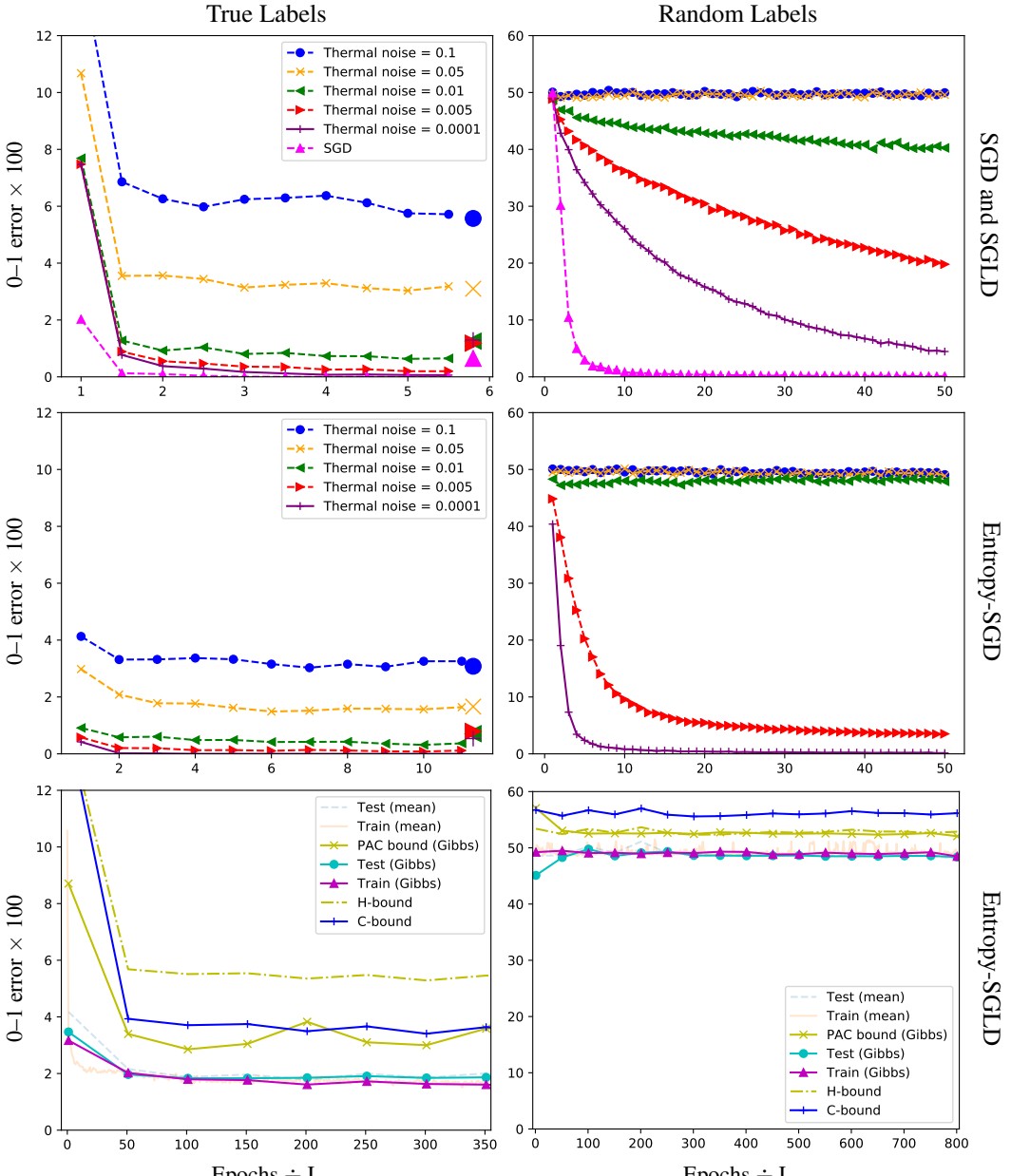

Figure 1: Results on the CONV network on two-class MNIST. **(top)** Training error (under 0–1 loss) for SGLD on the empirical risk $-\tau \hat{R}_S$ under a variety of thermal noise $\sqrt{2/\tau}$ settings. SGD corresponds to zero thermal noise. **(top-left)** The large markers on the right indicate test error. The gap is an estimate of the generalization error. On true labels, SGLD finds classifiers with relatively small generalization error. At low thermal noise settings, SGLD (and its zero limit, SGD), achieve small empirical risk. As we increase the thermal noise, the empirical 0–1 error increases, but the generalization error decreases. At 0.1 thermal noise, risk is close to 50%. **(top-right)** On random labels, SGLD has high generalization error for thermal noise values 0.01 and below. (True error is 50%). **(middle-left)** On true labels, Entropy-SGD, like SGD and SGLD, has small generalization error. For the same settings of thermal noise, empirical risk is lower. **(middle-right)** On random labels, Entropy-SGD overfits for thermal noise values 0.005 and below. Thermal noise 0.01 produces good performance on both true and random labels. **(bottom row)** Entropy-SGLD is configured to be $\varepsilon$-differentially private with $\varepsilon \approx 0.0327$ by setting $\tau = \sqrt{m}$, where $m$ is the number of training samples. **(bottom-left)** On true labels, the generalization error for networks learned by Entropy-SGLD is close to zero. Generalization bounds are relatively tight. **(bottom-right)** On random label, Entropy-SGLD does not overfit. See Fig. 3 for SGLD bounds at same privacy setting.

associated Gibbs posterior (i.e., a randomized classifier). We also report the performance of the (deterministic) network parametrized by these weights (called the "mean" classifier) in order to give a coarse statistic summarizing the local empirical risk surface.

Following Zhang et al. (2017), we study these algorithms on MNIST with its original ("true") labels, as well as on random labels. Parameter $\tau$ that performs very well in one setting often does not perform well in the other. Random labels mimic data where the Bayes error rate is high, and where overfitting can have severe consequences.

## 6.1 DETAILS

We use a two-class variant of MNIST (LeCun, Cortes, and Burges, 2010).[4] (See Fig. 4 and Appendix C for our experiments on the standard multiclass MNIST dataset. They yield similar insight.) Some experiments involve random labels, i.e., labels drawn independently and uniformly at random at the start of training. We study three network architectures, abbreviated FC600, FC1200, and CONV. Both FC600 and FC1200 are 3-layer fully connected networks, with 600 and 1200 units per hidden layer, respectively. CONV is a convolutional architecture. All three network architectures are taken from the MNIST experiments by Chaudhari et al. (2017), but adapted to our two-class version of MNIST.[5] Let $S$ and $S_{\text{tst}}$ denote the training and test sets, respectively. For all learning algorithms we track

(i) $\hat{R}_S(\mathbf{w})$ and $\hat{R}_{S_{\text{tst}}}(\mathbf{w})$, i.e., the training and test error for $\mathbf{w}$.

We also track

(ii) estimates of $\hat{R}_S(G_{\gamma,\tau}^{\mathbf{w},S})$ and $\hat{R}_{S_{\text{tst}}}(G_{\gamma,\tau}^{\mathbf{w},S})$, i.e., the mean training and test error of the local Gibbs distribution, viewed as a randomized classifier ("Gibbs")

and, using the differential privacy bounds in Theorem 5.5, compute

(iii) a PAC-Bayes bound on $R_{\mathscr{D}}(G_{\gamma,\tau}^{\mathbf{w},S})$ using Theorem 5.4 ("PAC-bound");

(iv) the mean of a Hoeffding-style bound on $R_{\mathscr{D}}(\mathbf{w}')$, where $\mathbf{w}' \sim P_{\exp(F_{\gamma,\tau}(\cdot;S))}$, using Theorem 5.2 ("H-bound");

(v) an upper bound on the mean of a Chernoff-style bound on $R_{\mathscr{D}}(\mathbf{w}')$, where $\mathbf{w}' \sim P_{\exp(F_{\gamma,\tau}(\cdot;S))}$, using Theorem 5.3 ("C-bound").

We also compute H- and C- bounds for SGLD, viewed as a sampler for $\mathbf{w}' \sim P_{\exp(-\tau\hat{R}_S)}$, where $P$ here is Lebesgue measure.

In order for SGLD and Entropy-SGLD to be private, we modify the cross entropy loss function to be bounded. We achieve this by an affine transformation of the neural networks output that prevents extreme probability (se Appendix B.2.1). With the choice of $\tau = \sqrt{m}$, and the loss function taking values in an interval of length $L_{\max} = 4$, Entropy-SGLD is $\varepsilon$-differentially private, with $\varepsilon \approx 0.0327$. See Appendix B.2 for additional details. Note that, in the calculation of (iii), we do not account for Monte Carlo error in our estimate of $\hat{R}_S(\mathbf{w})$. The effect is small, given the large number of iterations of SGLD performed for each point in the plot. Recall that

$$R_{\mathscr{D}}(G_{\gamma,\tau}^{\mathbf{w},S}) = \mathop{\mathbb{E}}_{\mathbf{w}' \sim G_{\gamma,\tau}^{\mathbf{w},S}}(R_{\mathscr{D}}(\mathbf{w}')), \qquad (24)$$

and so we may interpret the bounds in terms of the performance of a randomized classifier or the mean performance of a randomly chosen classifier.

---

[4] The MNIST handwritten digits dataset (LeCun, Cortes, and Burges, 2010) consists of 60000 training set images and 10000 test set images, labeled 0–9. We transformed MNIST to a two-class (i.e., binary) classification task by mapping digits 0–4 to label 1 and 5–9 to label $-1$.

[5] We adapt the code provided by Chaudhari et al., with some modifications to the training procedure and straightforward changes necessary for our binary classification task.

## 6.2 RESULTS

Key results for the convolutional architecture (CONV) appear in Fig. 1. Results for FC600 and FC1200 appear in Fig. 2 of Appendix B. (Training the CONV network produces the lowest training/test errors and tightest generalization bounds. Results and bounds for FC600 are nearly identical to those for FC1200, despite FC1200 having three times as many parameters.)

The top row of Fig. 1 presents the performance of SGLD for various levels of thermal noise $\sqrt{2/\tau}$ under both true and random labels. (Under our privacy approximation, we may also use SGLD to directly perform a private optimization of the empirical risk surface. The level of thermal noise determines the differential privacy of SGLD and so we expect to see a tradeoff between empirical risk and generalization error. Note that SGD is the same as SGLD with zero thermal noise.) SGD achieves the smallest training and test error on true labels, but overfits the worst on random labels. In comparison, SGLD's generalization performance improves with higher thermal noise, while its risk performance worsens. At 0.05 thermal noise, SGLD achieves reasonable but relatively large risk but almost zero generalization error on both true and random labels. Other thermal noise settings have either much worse risk or generalization performance.

The middle row of Fig. 1 presents the performance of Entropy-SGD for various levels of thermal noise $\sqrt{2/\tau}$ under both true and random labels. As with SGD, Entropy-SGD's generalization performance improves with higher thermal noise, while its risk performance worsens. At the same levels of thermal noise, Entropy-SGD outperforms the risk and generalization error of SGD. At 0.01 thermal noise, Entropy-SGD achieves good risk and low generalization error on both true and random labels. However, the test-set performance of Entropy-SGD at 0.01 thermal noise is still worse than that of SGD. Whether this difference is due to SGD overfitting to the MNIST test set is unclear and deserves further study.

The bottom row of Fig. 1 presents the performance of Entropy-SGLD with $\tau = \sqrt{m}$ on true and random labels. (This corresponds to approximately 0.09 thermal noise.) On true lables, both the mean and Gibbs classifier learned by Entropy-SGLD have approximately 2% test error and essentially zero generalization error, which is less than predicted by our bounds. Our PAC-Bayes risk bounds are roughly 3%. As expected by the theory, Entropy-SGLD does not overfit on random labels, even after thousands of epochs.

We find that our PAC-Bayes bounds are generally tighter than the H- and C-bounds. All bounds are nonvacuous, though still loose. The error bounds reported here are tighter than those reported by Dziugaite and Roy (2017). However, *the validity of all three privacy-based bounds that we report rests on the privacy approximation regarding SGLD*, and so interpreting these bounds requires some subtlety. We achieve much tighter generalization bounds than previously reported, and better test error, but we are still far from the performance of SGD. This is despite making a strong approximation, and so we might view these results as telling us the limits of combining differential privacy and PAC-Bayes bounds. Weaker notions of stability/privacy may be necessary to achieve further improvement in generalization error and test error. Despite the coarse privacy approximation, no bound is ever violated: possible explanations include the bounds simply being loose and/or the data being far from worst case. Note that, given the number of experiments, we might even expect a violation for tight bounds. Indeed, our performance on random labels supports the hypothesis that the privacy of (Entropy-)SGLD does not degrade over time, at least not in a way that can be detected by our experiments.

## 7 DISCUSSION

Our work reveals that Entropy-SGD can be understood as optimizing a PAC-Bayes generalization bound in terms of the bound's prior. Because the prior must be independent of the data, the bound is invalid, and, indeed, we observe overfitting in our experiments with Entropy-SGD when the thermal noise $\sqrt{2/\tau}$ is set to 0.0001 as suggested by Chaudhari et al. for MNIST.

PAC-Bayes priors can, however, depend on the data distribution. This flexibility seems wasted, since the data sample is typically viewed as one's only view onto the data distribution. However, using differential privacy, we can span this gap. By performing a private computation on the data, we can extract information about the underlying distribution, without undermining the statistical

validity of a subsequent PAC-Bayes bound. Our PAC-Bayes bound based on a differentially private prior is made looser by the use of a private data-dependent prior, but the gains in choosing a data-distribution-dependent prior more than make up for the expansion of the bound due to the privacy. (The gains come from the KL term being much smaller on the account of the prior being better matched to the posterior.) Understanding how our approach compares to local PAC-Bayes priors (Catoni, 2007) is an important open problem.

The most elegant way to make Entropy-SGD private is to replace SGD with a sample from the Gibbs distribution (known as the exponential mechanism in the differential privacy literature). However, generating an exact sample is intractable, and so practicioners use SGLD to generate an approximate sample, relying on the fact that SGLD converges weakly to the exponential mechanism under certain technical conditions. Our privacy approximation allows us to proceed with a theoretical analysis by assuming that SGLD achieves the same privacy as the exponential mechanism. On the one hand, we do not find overt evidence that our approximation is grossly violated. On the other, we likely do not require such strong privacy in order to control generalization error.

We might view our privacy-based bounds as being optimistic and representing the bounds we might be able to achieve rigorously should there be a major advance in private optimization. (No analysis of the privacy of SGLD takes advantage of the fact that it mixes weakly.) On the account of using private data-dependent priors, our bounds are significantly tighter than those reported by Dziugaite and Roy (2017). However, despite our bounds potentially being optimistic, the test set error we are able to achieve is still 5–10 times that of SGD. Differential privacy may be too conservative for our purposes, leading us to underfit. Indeed, we think it is unlikely that Entropy-SGD has strong differential privacy, yet we are able to achieve good generalization on both true and random labels under 0.01 thermal noise. Identifying the appropriate notion of privacy/stability to combine with PAC-Bayes bounds is an important problem.

Despite our progress on building learning algorithms with strong generalization performance, and identifying a path to much tighter PAC-Bayes bounds, Entropy-SGLD learns much more slowly than Entropy-SGD, the risk of Entropy-SGLD is far from state of the art, and our PAC-Bayes bounds are loose. It seems likely that there is a fundamental tradeoff between the speed of learning, the excess risk, and the ability to produce a certificate of one's generalization error via a rigorous bound. Characterizing the relationship between these quantities is an important open problem.

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

## A  BACKGROUND: DIFFERENTIAL PRIVACY

Here we formally define some of the differential privacy related terms used in the main text. (See (Dwork, 2006; Dwork and Roth, 2014) for more details.)

Let $U, U_1, U_2, \ldots$ be independent uniform $(0,1)$ random variables, independent also of any random variables introduced by $\mathbb{P}$ and $\mathbb{E}$, and let $\pi : \mathbb{N} \times [0,1] \to [0,1]$ satisfy $(\pi(1,U), \ldots, \pi(k,U)) \overset{d}{=} (U_1, \ldots, U_k)$ for all $k \in \mathbb{N}$. Write $\pi_k$ for $\pi(k, \cdot)$.

**Definition A.1.** A *randomized algorithm* $\mathscr{A}$ from $R$ to $T$, denoted $\mathscr{A} : R \rightsquigarrow T$, is a measurable map $\mathscr{A} : [0,1] \times R \to T$. Associated to $\mathscr{A}$ is a (measurable) collection of random variables $\{\mathscr{A}_r : r \in R\}$ that satisfy $\mathscr{A}_r = \mathscr{A}(U, r)$. When there is no risk of confusion, we will write $\mathscr{A}(r)$ for $\mathscr{A}_r$.

**Definition A.2.** A randomized algorithm $\mathscr{A} : Z^m \rightsquigarrow T$ is $(\varepsilon, \delta)$-*differentially private* if, for all pairs $S, S' \in Z^m$ that differ at only one coordinate, and all measurable subsets $B \subseteq T$, we have $\mathbb{P}(\mathscr{A}(S) \in B) \leq e^{\varepsilon} \mathbb{P}(\mathscr{A}(S') \in B) + \delta$.

We will write $\varepsilon$-differentially private to mean $(\varepsilon, 0)$-differentially private algorithm.

**Definition A.3.** Let $\mathscr{A} : R \rightsquigarrow T$ and $\mathscr{A}' : T \rightsquigarrow T'$. The *composition* $\mathscr{A}' \circ \mathscr{A} : R \rightsquigarrow T'$ is given by $(\mathscr{A}' \circ \mathscr{A})(u, r) = \mathscr{A}'(\pi_2(u), \mathscr{A}(\pi_1(u), r))$.

**Lemma A.4** (post-processing). *Let* $\mathscr{A} : Z^m \rightsquigarrow T$ *be* $(\varepsilon, \delta)$-*differentially private and let* $F : T \rightsquigarrow T'$ *be arbitrary. Then* $F \circ \mathscr{A}$ *is* $(\varepsilon, \delta)$-*differentially private.*

## B  TWO-CLASS MNIST EXPERIMENTS

### B.1  ARCHITECTURE

We studied three architectures: CONV, FC600, and FC1200.

CONV was a convolutional neural network, whose architecture was the same as that used by Chaudhari et al. (2017) for multiclass MNIST classification, except modified to produce a single probability output for our two-class variant of MNIST. In particular, CONV has two convolutional layers, a fully connected ReLU layer, and a sigmoidal output layer, yielding $126,711$ parameters in total.

FC600 and FC1200 are fully connected 3-layer neural networks, with 600 and 1200 hidden units, respectively, yielding $834,601$ and $2,385,185$ parameters in total, respectively. We used ReLU activations for all but the last layer, which was sigmoidal to produce an output in $[0,1]$.

In their MNIST experiments, Chaudhari et al. (2017) use dropout and batch normalization. We did not use dropout. The bounds we achieved with and without batch norm were very similar. Without batch norm, however, it was necessary to tune the learning rates. Understanding the combination of SGLD and batch norm and the limiting invariant distribution, if any, is an important open problem.

### B.2  TRAINING OBJECTIVE AND HYPERPARAMETERS FOR ENTROPY-SGLD

#### B.2.1  OBJECTIVE

All networks are trained to minimize a bounded variant of empirical cross entropy loss. The change involves replacing $g(p, y) = -\log p$ with $g(p, y) = -\log \psi(p)$, where

$$\psi(p) = e^{L_{\max}} + (1 - 2e^{-L_{\max}})p \tag{25}$$

is an affine transformation that maps to $[0,1]$ to $[e^{L_{\max}}, 1 - e^{L_{\max}}]$, removing extreme probability values. As a result, the binary cross entropy loss $\ell_{\mathrm{BCE}}$ is contained in an interval of length $L_{\max}$. In particular, $g(p, y) \in [0, L_{\max}]$. We take $L_{\max} = 4$ in our experiments.

#### B.2.2  EPOCHS

Ordinarily, an epoch implies one pass through the entire data set. For SGD, each stochastic gradient step processes a minibatch of size $K = 128$. Therefore, an epoch is $m/K = 468$ steps of SGD. An epoch for Entropy-SGD and Entropy-SGLD is defined as follows: each iteration of the inner SGLD

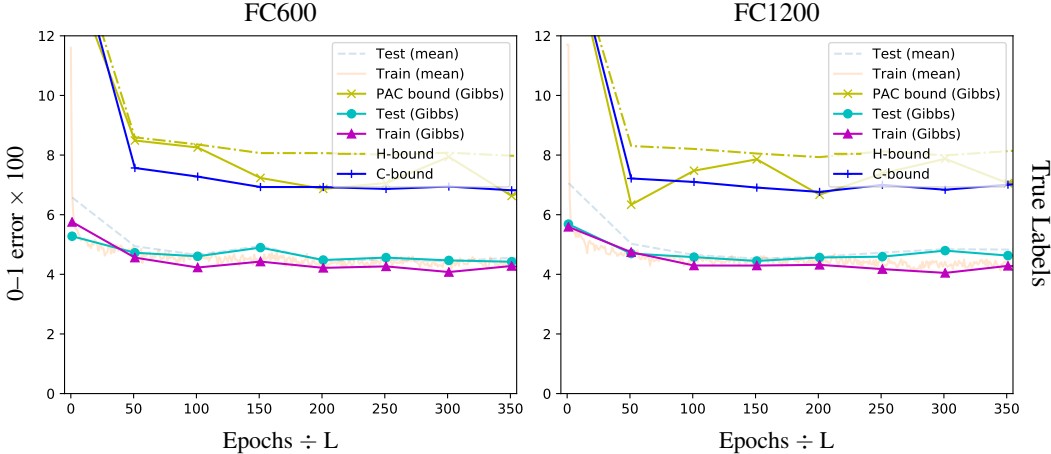

Figure 2: Fully connected networks trained on binarized MNIST with a differentially private Entropy-SGLD algorithm. **(left)** Entropy-SGLD applied to FC600 network trained on true labels. **(right)** Entropy-SGLD applied to FC1200 network trained on true labels. Both training error and generalization error are similar for both network architectures. T he true generalization gap is close to zero, since the test and train error overlaps. All the computed bounds on the test error are loose but nonvacuous.

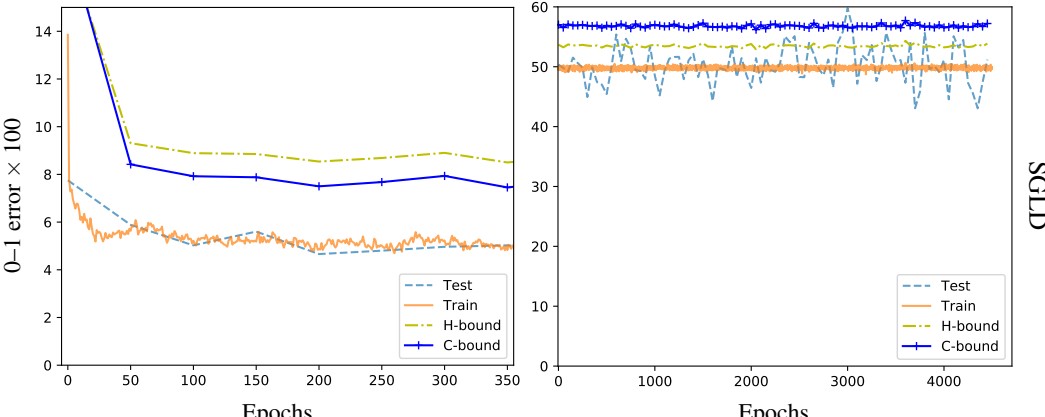

Figure 3: Results on CONV architecture, running SGLD configured to have the same differential privacy as Entropy-SGLD with $\tau = \sqrt{m}$. On true labels, SGLD learns a network with approximately 3% higher training and test error than the mean and Gibbs networks learned by Entropy-SGLD. SGLD does not overfit on random labels, as predicted by theory. The C-bound on the true error of this network is around 8%, which is worse than the roughly 4% C-bound on the mean classifier.

loop processes a minibatch of size $K = 128$, and the inner loop runs for $L = 20$ steps. Therefore, an epoch is $m/(LK)$ steps of the outer loop. In concrete terms, there are 20 steps of SGD per every one step of Entropy-SG(L)D. Concretely, the x-axis of our plots measure epochs divided by $L$. This choice, used also by Chaudhari et al. (2017), ensures that the wall-clock time of Entropy-SG(L)D and SGD align.

### B.2.3   SGLD parameters: step sizes and weighted averages

The step sizes for SGLD must be square summable but not summable. The step sizes for the outer SGLD loop are of the form $\eta_t = \eta t^{-0.6}$, with $\eta = \frac{0.006}{\gamma\tau}$. The step sizes for the inner SGLD loop are of the form $\eta_t = \eta t^{-1}$, with $\eta = \frac{1}{\gamma\tau}$.

The estimate produced by the inner SGLD loop is computed using a weighted average (line 8) with $\alpha = 0.75$. We use SGLD again when computing the PAC-Bayes generalization bound (Appendix B.3.2). In this case, SGLD is used to sample from the local Gibbs distribution when estimating the Gibbs risk and the KL term. We run SGLD for 1000 epochs to obtain our estimate. Again, we use weighted averages, but with $\alpha = 0.005$, in order to average over a larger number of samples and better control the variance.

### B.2.4   Gibbs classifier parameters

We set $\gamma = 1$ and $\tau = \sqrt{m}$ and keep the values fixed during optimization. By Theorem 5.5, the value of $\tau$, $L_{\max}$, and $\beta$ determine the differential privacy of Entropy-SGLD. In turn, the differential privacy parameter $\varepsilon$ and confidence parameter $\delta$ contribute

$$2\frac{\max\{\ln\frac{3}{\delta},\, m\varepsilon^2\}}{m-1} \tag{26}$$

to the bound on the KL-generalization error $\mathrm{KL}(\hat{R}_S(Q)\|R_\mathscr{D}(Q))$ in our differentially private PAC-Bayes bound (Theorem 5.4). Choosing $\tau = \sqrt{m}$, implies that the contribution coming from differential privacy decays at a rate of $1/m$. Numerically, given $L_{\max} = 4$ and $\beta = 1$, this contribution is 0.002.

### B.3   Evaluating the PAC-Bayes bound

### B.3.1   Inverting $\mathrm{KL}(q\|p)$

When the empirical error is close to zero, the KL version of the PAC-Bayes bound Theorem 3.1 is considerably tighter than the Hoeffding-style bound first described by McAllester (1999). However, using this relative entropy bound requires one to be able to compute the largest value $p$ such that $\mathrm{KL}(q\|p) \leq c$. There does not appear to be a simple formula for this value. In practice, however, the value can be efficiently numerically approximated using, e.g., Newton's method. See (Dziugaite and Roy, 2017, §2.2 and App. B).

### B.3.2   Estimating the KL divergence

Let $\ell(\mathbf{w}) = \tau\tilde{R}_S(\mathbf{w})$. By (Catoni, 2007, Lem. 1.1.3),

$$\mathrm{KL}(P_{\exp(-\ell)}\|P) = \mathop{\mathbb{E}}_{\mathbf{w}\sim P_{\exp(-\ell)}}[-\ell] - \log P[\exp(-\ell)]. \tag{27}$$

Both terms have obvious Monte Carlo estimates:

$$\mathop{\mathbb{E}}_{\mathbf{w}\sim P_{\exp(-\ell)}}[-\ell(\mathbf{w})] \approx -\frac{1}{k'}\sum_{i=1}^{k'}\ell(\mathbf{w}') \tag{28}$$

where $\mathbf{w}'_1,\ldots,\mathbf{w}'_{k'}$ are taken from a Markov chain targeting $P_{\exp(-\ell)}$, such as SGLD run for $k' \gg 1$ steps (which is how we computed our bounds), and

$$\log P[\exp(-\ell)] = \log \int \exp\{-\ell(\mathbf{w})\}P(\mathrm{d}\mathbf{w}) \tag{29}$$

$$\gtrsim \log \frac{1}{k}\sum_{i=1}^{k}\exp\{-\ell(\mathbf{w}_i)\}. \tag{30}$$

where $h_1,\ldots,h_k$ are i.i.d. $P$ (which is a multivariate Gaussian in this case). In the latter case, due to the concavity of log, the estimate is a lower bound with high probability, yielding a high probability upper bound on the KL term.

## C  MULTICLASS MNIST EXPERIMENTS

We evaluate the same generalization bounds on the standard MNIST classification task as in the MNIST binary labelling case. All the details of the network architectures and parameters are as stated in Appendix B.2, with two exception: following Chaudhari et al. (2017), we use a fully connected network with 1024 hidden units per layer, denoted FC1024.

### C.1  OBJECTIVE

The neural network produces a probability vector $(p_1,\ldots,p_K)$ via a soft-max operation. Ordinarily, we then apply the cross entropy loss corresponding to $g((p_1,\ldots,p_K),y) = -\log p_y$. When training privately, we use a bounded variant of the cross entropy loss, where the function $g$ above is replaced by $g((p_1,\ldots,p_K),y) = -\log \psi(p_y)$, and $\psi$ is defined as in Eq. (25).

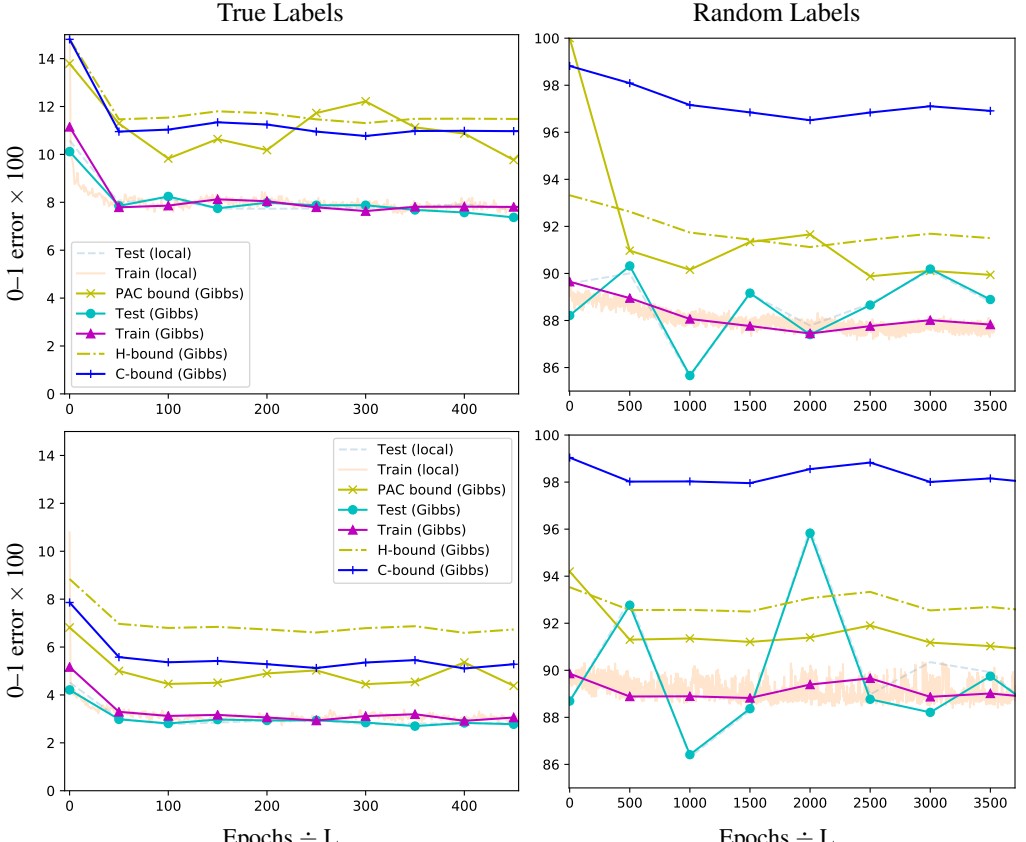

Figure 4: (⋆) Local here refers to the mean classifier. Entropy-SGLD results on MNIST. **(top-left)** FC1024 network trained on true labels. The train and test error suggest that the generalization gap is close to zero, while all three bounds exceed the test error by slightly more than 3%. **(bottom-left)** CONV network trained on true labels. Both the train and the test errors are lower than those achieved by the FC1024 network. We still do not observe overfitting. The C-bound and PAC-Bayes bounds exceed the test error by $\approx 3\%$. **(top-right)** FC1024 network trained on random labels. After approximately 1000 epochs, we notice overfitting by $\approx 2\%$. Running Entropy-SGLD further does not cause an additional overfitting. Theory suggests that our choice of $\tau$ prevents overfitting via differential privacy. **(bottom-right)** CONV network trained on random labels. We observe almost no overfitting (less than 1%). Both training and test error coincide and remain close to the guessing rate (90%).