# OpenReview forum: "Entropy-SGD optimizes the prior of a PAC-Bayes bound: Data-dependent PAC-Bayes priors via differential privacy"
_ICLR.cc/2018/Conference — Reject_

### Official Review · AnonReviewer2 · 2017-11-28

**Rating:** 6
**Confidence:** 3

**Review:**

This paper connects Entropy-SGD with PAC-Bayes learning. It shows that maximizing the local entropy during the execution of Entropy-SGD essentially minimize a PAC-Bayes bound on the risk of the Gibbs posterior. Despite this connection, Entropy-SGD could lead to dependence between prior and data and thus violate the requirement of PAC-Bayes theorem. The paper then proposes to use a differentially private prior to get a valid PAC-Bayes bound with SGLD. Experiments on MNIST shows such algorithm does generalize better.

Linking Entropy-SGD to PAC-Bayes learning and making use of differential privacy to improve generalization is quite interesting. However, I'm not sure if the ideas and techniques used to solve the problem are novel enough.
It would be better if the presentation of the paper is improved. The result in Section 4 can be presented in a theorem, and any related analysis can be put into the proof. Section 5 about previous work on differentially private posterior sampling and stability could follow other preliminaries in Section 2. The figures are a bit hard to read. Adding sub-captions and re-scaling y-axis might help.

---

> ### Author Response · Authors · 2017-12-21
> **Response**
>
> Thank you for your feedback.
>
> You raise two issues regarding novelty and clarity/presentation. We will address these in turns.
>
> Regarding novelty. We have recently presented this work to experts at a PAC-Bayes workshop. They expressed great interest in our results using differential privacy, and we have fielded a number of requests for preprints. Our private data-dependent priors can be viewed as a new type of data-distribution-dependent prior. The classical technique for dealing with data-distribution-dependent priors is due to Catoni and Lever et al., but these techniques have only been applied to Gibbs distributions, whereas our approaches offers much more flexibility. We now explain this connection more carefully in the related work section. We believe that our approach opens up the avenue to more advanced uses of stable, data-dependent priors.
>
> Beyond connecting PAC-Bayes theory and privacy, our work makes a number of other contributions:
> - We reveal the importance of the thermal noise to the generalization performance of Entropy-SGD, and tie this parameter to stability/privacy.  We also make a detailed study of the role of thermal noise in overfitting on MNIST, not only for Entropy-SGD, but also for SGLD and Entropy-SGLD.
> - We identify the deep connection between Entropy-SGD and PAC-Bayes bounds, which guides us to new ways to improve the generalization performance of Entropy-SGD. Our modifications lead to new learning algorithms that do not overfit, yet still have very good risk.
> - We obtain risk/generalization bounds for neural networks that, up to our privacy approximation, are much tighter than any bounds previously published for MNIST.
>
> Regarding clarity/presentation. We have rewritten several sections in the paper using your feedback as a guideline. Our connection between Entropy-SGD and PAC-Bayes priors is now stated as a theorem and our argument is now structured as a proof. Our derivations concerning privacy are now also organized into a theorem in Section 5. Indeed, Section 5 has been reworked from the ground up to have much clearer logical structure. We have reproduced all figures with larger fonts and careful attention to readability. The organization of Figure 1 now makes it immediately clear which figures are on true or random labels, and which algorithms are being compared.

---

> ### Author Response · Authors · 2018-01-19
> **Your issues have been addressed...**
>
> We revised our paper considerably over a month ago. We have since had a long back and forth conversation with AnonReviewer3 discussing the privacy approximation, which seems to have addressed their misgivings.
>
> We would much appreciate it if you could update your reviews and/or score.

---

### Official Review · AnonReviewer1 · 2017-11-28
**Weak Accept**

**Rating:** 6
**Confidence:** 3

**Review:**

1) I would like to ask for the clarification regarding the generalization guarantees. The original Entropy-SGD paper shows improved generalization over SGD using uniform stability, however the analysis of the authors rely on an unrealistic assumption regarding the eigenvalues of the Hessian (they are assumed to be away from zero, which is not true at least at local minima of interest). What is the enabling technique in this submission that avoids taking this assumption? (to clarify: the analysis is all-together different in both papers, however this aspect of the analysis is not fully clear to me).
2) It is unclear to me what are the unrealistic assumptions made in the paper. Please, list them all in one place in the paper and discuss in details.

---

> ### Author Response · Authors · 2017-12-21
> **Response to your questions**
>
> Thank you for your questions. We'll address both in turns, paraphrasing each as we understood it. We close with two remarks about uniform stability.
>
> 1. In our paper we repeat the statement by Chaudhari et al. that their analysis has some violated assumptions about curvature. How does our result sidestep this issue with the curvature.
>
> Our PAC-Bayes bound is tight provided that the KL(Q||P) term is small. In our case, P is a Gaussian whose mean is differentially private. Q is then the corresponding Gibbs posterior. Whether the empirical risk surface near the mean of P is exactly flat or nearly flat does not matter. In both cases, Q and P will be nearly identical and KL(Q||P) will be very small. This is what we find empirically.
>
> 2. What are the "unrealistic assumptions" you refer to?
>
> There is one approximation made in our paper: our "privacy approximation".  We now discuss this approximation in the Section 1, Introduction; Section 5, Data-dependent PAC-Bayes priors via differential privacy; Section 6, Numerical results on MNIST, and Section 7, Discussion.
>
> Our approximation is as follows (Section 1 and especially 5 give these details): The gold standard way to minimize a bounded function f is the exponential mechanism, namely generating a sample from the distribution with density exp(- c*f) where c > 0 is a constant. The bound on f and c determine the privacy. However, if f is high-dimensional and nonconvex, then exact sampling can be intractable. SGLD is a way to get an approximate sample, and it is know that the longer you run SGLD, the better the approximation. We approximate the exponential mechanism (i.e., an exact sample), with an approximate sample from SGLD, and calculate the privacy as if we got an exact sample. Differential privacy is a worst case framework and so we might not notice this approximation on "nice" data. An adversary might be able to exploit our approximation if they could carefully craft the data distribution. In the text, we point out that our bounds may be optimistic as a result, but they behave in a way that the theory predicts, and so we can still learn something from studying them.
>
> Finally, we'll make two remarks about the uniform stability of SGD and Entropy-SGD.
>
> First, the stability analysis in Chaudhari et al.'s Entropy-SGD paper does not account for the thermal noise required to get reasonable empirical results. Once you add in the amount of thermal noise they were advocating in their experiments, their results flips: Entropy-SGD is less stable. Our results actually point to using less thermal noise in order to get good generalization at the cost of excess empirical risk.
>
> Second, in the now well-known "Rethinking generalization" paper by Zhang, et al 2017, the authors, which include Hardt and Recht themselves, say that the uniform stability result cannot explain the difference between the performance on random and true labels, because stability does not care about the labels. The uniform stability bounds degrade to vacuous bouds after several passes through the data. The same issues are relevant to the stability analyses of Entropy-SGD.

---

> ### Author Response · Authors · 2018-01-19
> **The issues you've raised have been addressed.**
>
> Dear AnonReviewer1,
>
> We have addressed all your concerns.  We've also had a lengthy conversation with AnonReviewer3 around the privacy approximation. That reviewer appears to be now convinced of the reasonableness of our approximation.
>
> We would very much appreciate if you could update your reviews/scores.

---

### Official Review · AnonReviewer3 · 2017-12-06
**Reasonably good idea (but with lots of strong assumptions) connecting generalization of entropy SGD and PAC-Bayes risk bound.**

**Rating:** 6
**Confidence:** 3

**Review:**

Brief summary:
    Assume any neural net model with weights w. Assume a prior P on the weights. PAC-Bayes risk bound show that for ALL other distributions Q on the weights, the the sample risk (w.r.t to the samples in the data set) and expected risk (w.r.t distribution generating samples) of the random classifier chosen according to Q, averaged over Q, are close by a fudge factor that is KL divergence of P and Q scaled by m^{-1} + some constant.

Now, the authors first show that optimizing the objective of the Entropy SGD algorithm is equivalent to optimizing the empiricial risk term + fudge term over all data dependent priors P and the best Q for that prior. However, PAC-Bayes bound  holds only when P is NOT dependent on the data. So the authors invoke results from differential privacy to show that as long as the prior choosing mechanism in the optimization algorithm is differentially private with respect to data, differentially private priors can be substituted for valid PAC-Bayes bounds rectifying the issue. They show that when entrop SGD is implemented with pure gibbs sampling steps (as in Algorithm 3), the bounds hold.

Weakness that remains is that the gibbs sampling step in Entropy SGD (as in algo 3 in the appendix) is actually approximated by samples from SGLD that converges to this gibbs distribution when run for infinite hops. The authors leave this hole unsolved. But under the very strong sampling assumption, the bound holds. The authors do some experiments with MNIST to demonstrate that their bounds are not trivial.

Strengths:
  Simple connections between PAC-Bayes bound and entropy SGD objective is the first novelty. Invoking results from differential privacy for fixing the issue of validity of PAC-Bayes bound is the second novelty. Although technically the paper is not very deep, leveraging existing results (with strong assumptions) to show generalization properties of entropy-SGD is good.

Weakness:
  a) Obvious issue : that analysis assumes the strong gibbs sampling step.
  b) Experimental results are ok. I see that the bounds computed are non-vacuous. - but can the authors clarify what exactly they seek to justify ?
 c) Typos:
   Page 4 footnote "the local entropy should not be <with>.." - with is missing.
   Eq 14 typo - r(h) instead of e(h)
   Definition A.2 in appendix - must have S and S' in the inequality -both seem S.

d) Most important clarification: The way Thm 5.1, 5.2 and the exact gibbs sampling step connect with each other to produce Thm 6.1 is in Thm B.1. How do multiple calls on the same data sample do not degrade the loss ? Explanation is needed. Because the whole process of optimization in TRAIN with may steps is the final 'data dependent prior choosing mechanism' that has to be shown to be differentially private. Can the authors argue why the number of iterations of this does not matter at all ?? If I get run this long enough, and if I get several w's in the process (like step 8 repeated many times in algorithm 3) I should have more leakage about the data sample S intuitively right ?

e) The paper is unclear in many places. Intro could be better written to highlight the connection at the expression level of PAC-Bayes bound and entropy SGD objective and the subsequent fix using differentially private prior choosing mechanism to make the connection provably correct. Why are all the algorithms in the appendix on which the theorems are claimed in the paper ??

Final decision: I waver between 6 and 7 actually. However I am willing to upgrade to 7 if the authors can provide sound arguments to my above concerns.

---

> ### Author Response · Authors · 2017-12-21
> **Response to your comments**
>
> Thank you for the comments and pinpointing several typos.
>
> We have made an extensive rewrite to address the weaknesses you identified. We will respond to each of them, but in a different order.
>
> (e) and (c) Regarding clarity/presentation and typos.
>
> We have rewritten and rearranged much of the paper to improve the logical structure. We have also addressed all the typos.
>
> - Entropy-SGD and Entropy-SGLD are now presented in the main body of the paper as a single combined algorithm, with the one difference highlighted.
> - Our analysis of the idealized exponential mechanism (what you refer to as gibbs sampling) is now presented as Theorem 5.5, and its relationship to Entropy-SGLD is clearly laid out in the same section. We also discuss our privacy approximation here in depth.
> - Our result relating Entropy-SGD and PAC-Bayes bound optimization are now presented as Theorem 4.1.
> - Our argument establishing the differentially-private PAC-Bayes bound is now structured as a proof.
>
>
> (d) and (a). Regarding strong gibb sampling (i.e., the exponential mechanism and our "privacy approximation" regarding SGLD).
>
> We have updated this part of the paper considerably, and the logical structure is much improved. The material is now entirely in the main body. We highlight some aspects of the argument here:
>
> - Note that we only use a SINGLE sample produced by SGLD (namely the last one). This last sample is what is used as the prior mean to produce the resulting Gibbs posterior classifier. When we plot the learning curves, the bounds are the bounds that would hold if we stopped SGLD at that iteration.
> - The fact we only use one sample is the reason why we think it is reasonable to approximate the privacy of SGLD by that of its limiting invariant distribution (i.e., the exponential mechanism). Since we are far from the worst case with MNIST, we expect not to see much difference. There is likely a worst-case distribution where our bounds would end up being badly violated.
> - Typical analyses of SGLD don't try to deal with the fact that it begins to mix. So they make a step by step analysis, where information is leaked at every stage. Because they do this, there is no reason not to release the whole trajectory. However, in an analysis that took advantage of mixing (very hard!), they would NOT release the whole trajectory (or at least, they certainly wouldn't release the early parts).
> - In our experiments where we run SGLD for 1000's of epochs (!) on random noise, we see zero overfitting when we set the thermal noise to the settings suggested by theory.
>
>
> (b) Regarding the goal of the experiments.
>
> We have significantly revised the section describing our numerical experiments. We feel that the motivation for our experiments in much clearer now. Here are some particular points we wanted to highlight:
>
> PAC-Bayes bounds are data dependent and so it is an empirical question whether they are useful or not, and how they compare to previously established bounds. On top of this, we are using private data-dependent priors and a differentially private PAC-Bayes theorem and so it is an empirical question whether a sufficiently private optimization finds a decent prior. (Generalization bounds require a very high degree of privacy!) One way to think about the quantity tau/m (which determines the privacy along with our loss bound) is that, when tau/m < 1, it specifies what fraction of your data you "throw away" while doing your sampling in order to not learn "too much" about your data itself, rather than the distribution underlying it. We have to "throw away" quite a bit of data while privately optimizing our PAC-bayes prior. And so it is an empirical question whether we can find anything useful still. Indeed, we do. We can also study our privacy approximation regarding SGLD empirically. If the privacy/stability of SGLD degraded over time, we might have seen overfitting occur on very long runs. In fact, we don't see this, even after 1000's of epochs! The private versions of the algorithms we tested reach some level of performance and stay there.

---

### Author Response · Authors · 2017-12-21
**Summary of the major changes we made addressing reviewer feedback**

This comment summarizes the major changes we made to the document while addressing the reviewers' comments. We have also crafted responses to each individual reviewer.

We took all of the reviewers’ comments seriously and made extensive edits to the article. Some of the major changes include:

1. stating our main results as theorems and writing up the analysis in the form of a proof. This should make our contributions clearer to readers. These results include: i) the connection between Entropy-SGD optimization and PAC-Bayes prior optimization, ii) our differentially private PAC-Bayes bound, iii) our privacy analysis for the data-dependent prior.

2. giving a single unified description of the Entropy-SGD and Entropy-SGLD algorithms, so the difference is obvious.

3. rewriting our differential privacy analysis, to make it easier for the reader to understand our assumptions/approximations.

3. adding experiments comparing SGLD and Entropy-SGD at different levels of thermal noise, which highlights the role of thermal noise in generalization and the difference between empirical risk minimization and local entropy maximization.

4. discussing the relationship between our differentially private PAC-Bayes priors and data-distribution-dependent priors.

---

### Decision · Program_Chairs · 2018-01-29
**ICLR 2018 Conference Acceptance Decision**

**Decision:**

Reject

**Comment:**

The paper proposes a new analysis of the optimization method called entropy-sgd which seemingly leads to more robust neural network classifiers. This is a very important problem if successful. The reviewers are on the fence with this paper. On the one hand they appreciate the direction and theoretical contribution, while on the other they feel the assumptions are not clearly elucidated or justified. This is important for such a paper. The author responses have not helped in alleviating these concerns. As one of the reviewers points out, the writing needs a massive overhaul. I would suggest the authors clearly state their assumptions and corresponding justifications in future submissions of this work.